# A Quantitative Phytochemical Comparison of Olive Leaf Extracts on the Australian Market

**DOI:** 10.3390/molecules25184099

**Published:** 2020-09-08

**Authors:** Ian Breakspear, Claudia Guillaume

**Affiliations:** 1Department of Naturopathic Medicine, Endeavour College of Natural Health, Sydney 2000, Australia; 2WholMed Consultancy, Sydney 2000, Australia; 3Modern Olives, Lara 3212, Australia; c.guillaume@modernolives.com.au

**Keywords:** *Olea europaea*, olive leaf extract, OLE, olive leaf, oleuropein, hydroxytyrosol, phytoequivalence, herbal medicine, phytomedicine

## Abstract

Olive leaf extract (OLE), prepared from the fresh or dried leaves of *Olea europaea* L., is generating interest as a cardiovascular and metabolic disease risk modifier. Positive effects for the leaf extract and its key phytochemical constituents have been reported on blood pressure, respiratory infections, inflammation, and insulin resistance. A variety of OLE products are available both over-the-counter and for professional dispensing. The aim of this research was to quantitatively explore the phytochemical profile of different OLE products on the Australian market. Ten OLE products available on the Australian market (five over-the-counter products and five products for professional compounding and dispensing) were quantitatively analyzed for oleuropein, hydroxytyrosol, oleacein, oleocanthal, total biophenols, maslinic acid, and oleanolic acid, using high-performance liquid chromatography (HPLC). Substantial variation in oleuropein and hydroxytyrosol levels was noted between extracts, with a trend towards higher oleuropein and lower hydroxytyrosol levels being noted in products produced using the fresh olive leaf as opposed to dry olive leaf. These results suggest that OLE products on the Australian market vary substantially in their phytochemical profiles. Products for professional compounding and dispensing in many cases contained less oleuropein than over-the-counter products, but more hydroxytyrosol and comparable total biophenol levels.

## 1. Introduction

The leaves of the olive tree, *Olea europaea* L., have been used medicinally for more than 200 years. Daniel Hanbury, in 1854, quoted a passage from a letter he received from Sidney H. Maltass, in which he discussed the use of a decoction of the olive leaf as a substitute for quinine in the management of fever. Hanbury then continued to discuss the French use of the leaf in fever, as documented by M. Cazals in 1811, and the 1828 report by Dr. E. Pallas stating that Spanish physicians were also using the leaves as a febrifuge, and French military physicians were using it as a substitute for cinchona bark during the Spanish war in 1808–1813 [1].

In more recent times, olive leaf extract (OLE) and some of its key phytochemical constituents have demonstrated antioxidant, anti-inflammatory, anti-atherogenic, and antimicrobial activities [2,3,4,5]. Small scale human studies on OLE demonstrate promising effects on cardiovascular and metabolic disease risk factors, including hypertension [6,7] and insulin resistance [8], as well as respiratory infections [9] and oral mucositis in cancer chemotherapy patients [10].

Key active phytochemicals reported in OLE are mostly biophenols, including oleuropein, hydroxytyrosol, as well the pentacyclic triterpenoids—maslinic acid and oleanolic acid [11].

In Australia, OLE is available in a range of therapeutic products available over-the-counter in pharmacies and health food stores. Additionally, OLE can be professionally recommended by clinicians, including naturopaths, herbalists, and nutritionists, with some OLE products being specifically designed for extemporaneous compounding and dispensing by herbalists and naturopaths. Therapeutic indications for herbal products in Australia are derived from both traditional usage as well as scientific research [12]; however, medicinal plant extracts can show natural phytochemical variability based on factors such as growing conditions, harvest times, and extraction methods [13], which has the potential to alter efficacy [14].

End-users of olive leaf extract in Australia—whether they be consumers or professionals dispensing to their patients—have little information on which to base their choice of products, other than specifications provided on product labels. These label specifications may have limited utility due to the fact that some products declare quantities of specific phytochemicals in the extract, and some products do not. Thus a deeper understanding of the comparative concentrations of key olive leaf phytochemicals between products would be desirable. Therefore, the objective of this study was to quantitatively compare various olive leaf extract (OLE) products on the Australian market, specifically the key phytochemical markers of oleuropein, hydroxytyrosol, oleacein, oleocanthal, total biophenols, maslinic acid, and oleanolic acid.

## 2. Results

Substantial phytochemical variation was observed across the 10 extracts analyzed. Table 1 provides the concentration of oleuropein, hydroxytyrosol, oleacein, oleocanthal, total biophenols, maslinic acid, and oleanolic acid in each extract, expressed as parts per million and milligrams per milliliter.

### 2.1. Oleuropein, Hydroxytyrosol, and Total Biophenols—Comparison between Product Categories

Figure 1 compares the concentration of oleuropein, hydroxytyrosol, and total biophenols in parts per million for all products.

For oleuropein, across the 10 extracts sampled, the results ranged between a minimum of 383.60 ppm to a maximum of 11,653.72 ppm, equating to a 30.4-fold variation in oleuropein levels. On comparing over-the-counter (OTC) products and products for professional compounding and dispensing (PROF), the mean oleuropein concentration for OTC products was 6680.30 ppm, and for PROF products, it was 2749.54 ppm, which equated to a 2.4-fold variation.

For hydroxytyrosol, across the 10 extracts sampled, the results ranged between a minimum of 163.10 ppm to a maximum of 6215.90 ppm, equating to a 38.1-fold variation in hydroxytyrosol levels. On comparing OTC and PROF products, the mean hydroxytyrosol concentration for OTC products was 405.58 ppm, and for PROF products, it was 2772.20 ppm, which equated to a 6.8-fold variation.

For total biophenols, across the 10 extracts sampled, the results ranged between a minimum of 2744.60 ppm to a maximum of 12,229.00 ppm, equating to a 4.5-fold variation in total biophenol levels. On comparing OTC and PROF products, the mean total biophenol concentration for OTC products was 6868.28 ppm, and for PROF products, it was 8669.68 ppm, which equated to a 1.3-fold variation.

### 2.2. Oleuropein, Hydroxytyrosol, and Total Biophenols—Comparison between Extracts Prepared from Fresh Leaf or Dry Leaf

All products had label declarations as to whether fresh or dry leaf was used for extraction. This allowed for the results to be compared between the extracts made from fresh leaf versus the extracts made from the dry leaf. It was observed that for the five OTC products sampled, four utilized the fresh leaf, and only one utilized dry leaf as raw material. However, for the five PROF products samples, four utilized the dry leaf, and one utilized the fresh leaf as raw material.

Figure 2 illustrates the ranges and means for oleuropein, hydroxytyrosol, and total biophenol concentrations, in fresh and dry leaf extracts.

### 2.3. Oleacein and Oleocanthal—Comparison between Product Categories

For oleacein, across the 10 extracts sampled, the results ranged between a minimum of 39.70 ppm to a maximum of 1773.30 ppm, equating to a 46.7-fold variation. On comparing OTC and PROF products, the mean oleacein concentration for OTC products was 593.68 ppm, and for PROF products, it was 215.54 ppm, which equated to a 2.8-fold variation.

For oleocanthal, only two products had detectable levels, with both products being OTC products from the same company.

### 2.4. Maslinic Acid and Oleanolic Acid—Comparison between Product Categories

For maslinic acid, across the 10 extracts sampled, the results ranged between a minimum of 0 ppm to a maximum of 813.96 ppm. On comparing OTC and PROF products, the mean maslinic acid concentration for OTC products was 45.63 ppm, and for PROF products, it was 254.19 ppm, which equated to a 5.8-fold variation.

For oleanolic acid, across the 10 extracts sampled, the results ranged between a minimum of 0 ppm to a maximum of 584.31 ppm. On comparing OTC and PROF products, the mean oleanolic acid concentration for OTC products was 183.91 ppm, and for PROF products, it was 244.05 ppm, which equated to a 1.3-fold variation.

### 2.5. Label Declarations of Oleuropein and Hydroxytyrosol and Quantity Per Daily Maximum Dosage

The strength of individual herbal products sold in Australia is declared on labels as the equivalent dry or fresh herb weight in the final product. For liquid extracts for professional extemporaneous dispensing, this is often achieved through a statement of the drug/extract ratio, expressed as the quantity of herbal material (fresh or dry) by weight, to final extract quantity (for liquid extracts, this is by volume). In some cases, additional quantitative statements may be made for specific phytochemicals in the finished product.

For the OLE products sampled, which made a quantitative statement regarding one or more phytochemicals, all of them met or exceeded their label declarations. However, it was noted that while all OTC products made a quantitative statement about one or more phytochemicals, none of the PROF products made such statements.

All products sampled had adult dosage ranges declared on their labels, providing an opportunity to compare quantities of key phytochemicals per dosage across the various products. Substantial variation in adult maximum daily dosage was noted between the products, with OTC products having higher daily dosage recommendations than most PROF products.

Table 2 summarizes the label information provided on all 10 OLE products, the maximum dosage recommendations for each product, and the corresponding results of quantification for oleuropein and hydroxytyrosol. Where the drug/extract ratio was not specifically stated on the label, it was calculated based on the information provided.

## 3. Discussion

As far as the authors are aware, this is the first published research, quantitatively comparing both over-the-counter liquid olive leaf extracts and liquid olive leaf extracts for extemporaneous dispensing by health professionals in Australia.

Substantial variation in the phytochemical profiles of OLE products was observed in this research, and in addition, it was discovered that some products were extracted from fresh olive leaf, while for other products, the olive leaf was dried prior to extraction.

A noticeable trend towards higher oleuropein and lower hydroxytyrosol levels was observed in the extracts made from fresh olive leaf compared to extracts made from dry olive leaf. Despite this trend, mean total biophenol levels were largely comparable between fresh and dry leaf extracts. This observed trend may be due to the stability issues with oleuropein, which can degrade to produce hydroxytyrosol [15]. While the drying process of the leaves may be a contributing factor, recent research of dried olive leaves grown in China has shown that oleuropein is still quantitatively greater than hydroxytyrosol [16]. Research has also demonstrated that certain yeasts isolated from black olive pomace have the potential to mediate the conversion of oleuropein to hydroxytyrosol under experimental conditions [17]; however, the relevance of this in commercial olive leaf extracts remains to be determined, given the far higher concentration of polyphenols and their inhibitory effect on microorganism growth.

While all over-the-counter products made quantitative label declarations for either oleuropein or hydroxytyrosol or both, none of the professional products made any such declaration. Given the context of dispensing of these professional products, whereby a health professional will make an individual assessment of a patient’s health status and subsequently customize the extemporaneous herbal formulation components and dosage, it is of concern that quantitative information regarding key phytochemical constituents was not provided.

This issue is compounded by the fact that recent human studies on OLE for blood pressure [6,7], insulin sensitivity [8], and respiratory illness [9] have utilized daily dosages of OLE, providing between 51 and 136 mg of oleuropein, with the studies assessing cardiovascular effects being on interventions containing 100 mg or more per day. At the maximum recommended dosage of each product, none of the professional products provided 51 mg of oleuropein per day, whereas only one OTC product was below 51 mg per day, and two of the five OTC products contained more than 100 mg oleuropein per day.

A key limitation of this research is that only one batch of each product was analyzed, and it is possible that batch to batch variation may be present and may reveal other trends. Consistency of herbal products from batch to batch is a known issue [14], so further investigations on multiple batches of olive leaf extract products would provide valuable insight.

In conclusion, this research demonstrates that there is substantial quantitative phytochemical variation between different olive leaf extracts on the Australian market. Additionally, while other factors are also likely to influence oleuropein levels in OLE products, manufacturers looking to maximize levels should probably consider utilizing fresh olive leaves rather than dry olive leaves in their extraction processes.

## 4. Materials and Methods

A total of 10 different liquid olive leaf extract (OLE) products available on the Australian marketplace were chosen for this study. Five of these products were available over-the-counter to the general public (designated OTC), and five products were liquid extracts sold to health professionals for extemporaneous compounding and dispensing (designated PROF). Only liquid OLE preparations were chosen to avoid confounding factors, which may be introduced through the process of creating tableted or encapsulated products, and to ensure comparability across the different product categories (as extracts for professional extemporaneous compounding were only available in liquid form). Specifications of these products are found in Table 3. Only unflavored liquid extracts were chosen for this investigation to avoid the risk of a flavoring compound altering the phytochemical profile. Other than extraction solvents (water, ethanol, and glycerol), the products did not declare the addition of any other additives.

The testing was conducted in August to September 2018 at the Modern Olives laboratory, in Lara, Victoria, Australia. The Modern Olives laboratory holds ISO 17025 accreditation (NATA #15594) and TGA license Nº: MI-2017-LI-01351-1.

### 4.1. Total Biophenols and Hydroxytyrosol Content

The International Olive Council method for the determination of biophenols was employed (COI/T.20/Doc No 29/Rev.1—2017). Samples were analyzed in high-performance liquid chromatography (HPLC) with DAD detection. In this method, the biophenolic compounds were extracted from OLE using methanol:water (80:20 *v*/*v*) solution. The subsequent quantification was by HPLC with the aid of a UV detector at 280 nm, and the contents of total biophenols and hydroxytyrosol were expressed in mg/kg. Apigenin was used as an internal standard.

### 4.2. Oleuropein

The European Pharmacopoeia method was employed for the quantification of oleuropein (Ph. Eur. monograph 2313 (2.2.29) IV-333—2018). The samples were prepared by weighing 0.1 g of OLE and adding 25 mL of methanol, mixing thoroughly, and then sonicating for 15 min. The content was then poured into a 50 mL flask and diluted with water to the mark. The sample was then analyzed by injecting 20 microliters into the HPLC, using a Licrospher C18, 5 micrometers (250 × 4.6 mm inner diameter) column, and UV detector at 233 nm. The mobile phase employed was trifluoroacetic acid:methanol:water in a ratio of 1:400:600. A calibration curve was used for the quantification, and the results were expressed in mg/kg, with the formula employed being:Oleuropein content (ppm) = A1 × V/C × mL(1)

A1 = area of oleuropeinmL = sample weight (g)V = final volume (mL)C = calibration curve gradient

### 4.3. Maslinic and Oleanolic Acid

For the analysis of triterpenic acids, samples were prepared by combining 1 g of OLE with 20 mL of ethanol in a 50 mL flask, then transferring to an ultrasonic bath for 30 min, then centrifuging at 4000 rpm for 10 min. After this extraction, the supernatant was collected into a flask, and the solvent evaporated to dryness. The residue was then dissolved in 10 mL of methanol and analyzed by HPLC using a UV detector at 210 nm. A calibration curve was used for the quantification of each acid, and the results were expressed in mg/kg. For the calibration curve, oleanolic acid and maslinic acid (from Sigma-Aldrich, St. Louis, MO, USA) were used as external standards with 6 different concentrations from 50 to 3000 mg/L, dissolved in methanol. The formula used is below:Maslinic acid (mg/kg) = AMA × V/CCMA × W(2)
Oleanolic acid (mg/kg) = AOA × V/CCOA × W(3)

AMA = area of maslinic acidAOA = area of oleanolic acidCCMA = calibration curve gradient of maslinic acid standardCCOA = calibration curve gradient of oleanolic acid standardV = final volume (mL)W = sample weight (g)

### 4.4. Density

The density of samples was determined by the European Pharmacopoeia weight method (Ph. Eur. method 2.2.5 V-A252 Appendix V G—2018).

## Figures and Tables

**Figure 1 molecules-25-04099-f001:**
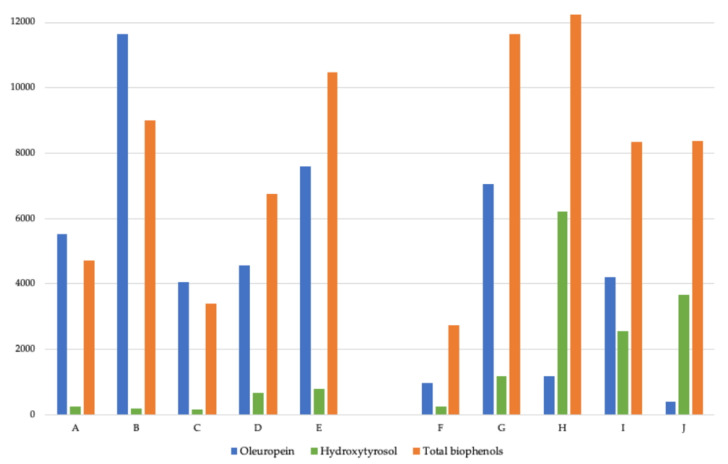
Comparison of oleuropein, hydroxytyrosol, and total biophenol quantities in parts per million. The grouping on the left (samples A–E) are over-the-counter products (OTC), and the grouping on the right (samples F–J) are products for professional extemporaneous compounding and dispensing (PROF).

**Figure 2 molecules-25-04099-f002:**
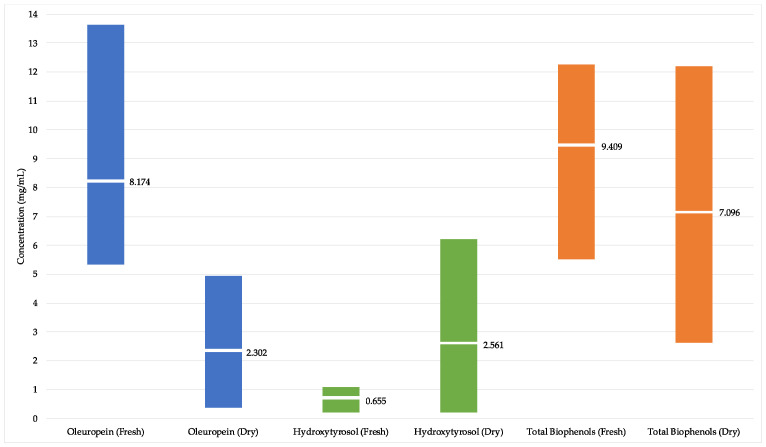
Ranges and mean concentrations of oleuropein, hydroxytyrosol, and total biophenols in extracts made from fresh leaf compared to the dry leaf. The vertical colored bars represent the ranges, and the horizontal labeled white bars represent the means.

**Table 1 molecules-25-04099-t001:** The concentration of phytochemicals quantified in each of the 10 extracts analyzed.

Sample	A	B	C	D	E	F	G	H	I	J
Density (g/mL)	1.170	1.170	1.210	1.170	1.170	0.952	0.929	0.997	0.987	0.996
Oleuropein (ppm)	5526.42	11653.72	4066.34	4560.88	7594.15	951.49	7050.89	1157.95	4203.78	383.60
Oleuropein (mg/mL)	6.466	13.635	4.920	5.336	8.885	0.906	6.550	1.154	4.149	0.382
Hydroxytyrosol (ppm)	241.20	174.30	163.10	667.40	781.90	236.90	1176.40	6215.90	2564.30	3667.50
Hydroxytyrosol (mg/mL)	0.282	0.204	0.197	0.781	0.915	0.226	1.093	6.197	2.531	3.653
Oleacein (ppm)	39.70	112.50	126.10	916.80	1773.30	103.60	254.50	305.80	245.20	168.60
Oleacein (mg/mL)	0.046	0.132	0.153	1.073	2.075	0.099	0.236	0.305	0.242	0.168
Oleocanthal (ppm)	0	0	0	41.70	59.70	0	0	0	0	0
Oleocanthal (mg/mL)	0	0	0	0.049	0.070	0	0	0	0	0
Total biophenols (ppm)	4719.40	8997.50	3380.70	6762.40	10481.40	2744.60	11647.10	12229.00	8352.90	8374.80
Total biophenols (mg/mL)	5.522	10.527	4.091	7.912	12.263	2.613	10.820	12.192	8.244	8.341
Maslinic acid (ppm)	0	7.45	9.51	66.98	144.21	274.82	813.96	131.93	21.26	28.98
Maslinic acid (mg/mL)	0	0.009	0.012	0.078	0.169	0.262	0.756	0.132	0.021	0.029
Oleanolic acid (ppm)	0	8.88	0	326.37	584.31	352.56	560.12	228.29	37.71	41.55
Oleanolic acid (mg/mL)	0	0.010	0	0.382	0.684	0.336	0.520	0.228	0.037	0.041

**Table 2 molecules-25-04099-t002:** Label declarations of sampled products and the tested quantity of phytochemicals per maximum adult daily dosage.

Sample	Summary of Label Declaration	DER *	Maximum Adult Daily Dosage (mL)	Oleuropein Per Maximum Dosage (mg)	Hydroxytyrosol Per Maximum Dosage (mg)
A	Each 15 mL contains leaf extract equiv approx 12 g fresh leaf, standardized to contain 66 mg oleuropein and 0.75 mg hydroxytyrosol	1:1.25 (fresh)	15	96.989	4.233
B	Each 10 mL contains extract equiv 8.5 g fresh leaf, standardized to contain 136 mg oleuropein	1:1.18 (fresh)	10	136.349	2.039
C	Each 5 mL contains extract equiv dry leaf 5 g, standardized oleuropein 22 mg	1:1 (dry)	10	49.203	1.974
D	Each 15 mL contains extract equiv 15 g fresh leaf, standardized to 75 mg oleuropein	1:1 (fresh)	15	80.043	11.713
E	Each 15 mL contains extract equiv 15 g fresh leaf, standardized to 110 mg oleuropein	1:1 (fresh)	15	133.277	13.722
F	n/a	1:2 (dry)	12	10.870	2.706
G	n/a	1:2 (fresh)	7	45.852	7.650
H	n/a	1:1 (dry)	3	3.463	18.592
I	Each 1 mL contains equiv 500 mg dry leaf	1:2 (dry)	7	29.044	17.717
J	Each 1 mL contains equiv 500 mg dry leaf	1:2 (dry)	7	2.674	25.570

* DER is drug/extract ratio, declared as weight in kilograms of fresh or dry herbal material to volume in liters of the finished liquid extract. If DER was not provided on the label, it was calculated from the quantitative statement found on the label.

**Table 3 molecules-25-04099-t003:** Specifications of the products analyzed.

Sample Code	Product Name	Batch	Expiry Date	Fresh or Dry Leaf	Product Category *	ARTG Identifier **
A	Comvita Medi Olive 66	15446	Sep 2019	Fresh	OTC	252958
B	Comvita Cardiovascular Support Medi Olive 136	15479	28 Sep 2019	Fresh	OTC	291745
C	Healthy Care Olive Leaf Extract	699984	Aug 2020	Dry	OTC	162092
D	Wellgrove Immune Support	OLE230218 (I)	31 Aug 2020	Fresh	OTC	312527
E	Wellgrove Heart Health	OLE60.8022	31 Jul 2020	Fresh	OTC	312833
F	Pharmaceutical Plant Company (dry)	130005	Jan 2020	Dry	PROF	n/a
G	Pharmaceutical Plant Company (fresh)	4512	Mar 2023	Fresh	PROF	n/a
H	Herbal Extract Company	280.140H7	Aug 2020	Dry	PROF	n/a
I	MediHerb	22760	Nov 2021	Dry	PROF	n/a
J	Optimal Rx	14273	9 Feb 2021	Dry	PROF	n/a

* OTC is available to the general public over-the-counter, PROF is available to health professionals for extemporaneous compounding and dispensing. ** ARTG Identifier is the Australian Register of Therapeutic Goods unique product identifier provided for finished therapeutic products sold on the Australian marketplace. Products for extemporaneous dispensing and compounding are considered raw materials and thus do not need to be included on the ARTG.

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
