# Peer review of "A Quantitative Phytochemical Comparison of Olive Leaf Extracts on the Australian Market"

_molecules, 2020, doi:10.3390/molecules25184099_

Round 1

Reviewer 1 Report

The paper deals with the quantification of phytochemical profile of various OLE products on the Australian market. The paper is well written and provided advancement in scientific knowledge, although the article could be improved along the following guidelines:

Page 3 ligne 82: The extraction method should be briefly described.

Page 3 ligne 86-87: The extraction method should be briefly described.

Page 3 ligne 87-88: How calibration curves were obtained? What is the standard used? Also, add the equation of each calibration curve.

A comparison of the obtained results with those of literature is strongly recommended. For example see: Journal of Microbiological Methods 176 (2020) 106010; Food Research International 128 (2020) 108785; Food Chemistry 279 (2019) 40-48

Reviewer 2 Report

This is an excellent study in which the authors were the first to examine  olive leaf extracts from fresh and dry leaves for bioactives associated with beneficial health properties. Ten different liquid preparations  available  in the Australia, five over the counter and 5 sold to  health professionals were studied.

The overall results were  nicely summarized  in the abstract. The introduction was short but covered  the medicinal properties  of  olive tree leaves first reported  almost 200 years ago followed by a brief review of their therapeutic properties.

The methods section was nicely detailed  with the results clearly shown and discussed. The data  presented is new and show the variability in phytochemical profiles in the different olive leave extracts particularly with respect to levels of oleuropein anf hydroxytyrosol. of particular interest was  the lower amounts of oleuropein in  the professional pounding  products  compared to the over the counter products. 

Author Response

No changes were suggested by Reviewer 2.

Reviewer 3 Report

Authors describing the objective of their research should indicate the novelty elements of the prepared manuscript. In addition, the phrase "The objective of this study was to quantitatively compare various products OLE". requires clarification.

Line 39: Authors pointing to a number of properties that show the extracts of the leaves of the olive tree should provide additional references.

Line 56: Why was only one series of purchased products analyzed? Were only 10 products available on the market? Did the purchased OLE products contain other chemical additives on the label? What does mean the expression "liquid OLE"? Please explain.

Line 72: A short description of the total biophenols determination should be presented by Authors.

Line 76: Authors extracted "the biophenolic compounds" from OLE using "methanol solution". Please enter its concentration.

Line 81: Please provide more information on the type of device (HPLC) used for the determination, eg oleuropein. What type of mobile phase was used in this study?

Line 86: Did Authors also use ultrasonic assisted extraction method to determine maslinic and oleanolic acids?

Figure 1 requires additional legend for hydroxytyrosol and total biophenols.

References: Latin plant names should be in italics (position 1, 3, 4).

Round 2

Reviewer 3 Report

After analyzing the answers on my remarks given by the authors I have no additional comments. I accept the applied changes in the corrected manuscript.

This manuscript is a resubmission of an earlier submission. The following is a list of the peer review reports and author responses from that submission.